# Phage Display’s Prospects for Early Diagnosis of Prostate Cancer

**DOI:** 10.3390/v16020277

**Published:** 2024-02-10

**Authors:** Valery A. Petrenko

**Affiliations:** Department of Pathobiology, College of Veterinary Medicine, Auburn University, Auburn, AL 36849, USA; petreva@auburn.edu

**Keywords:** phage display, landscape phage, molecular evolution, affinity selection, recombinant antibodies, PC, prostate-specific antigen (PSA), prostate-specific matrix antigen (PSMA), enzyme-linked immunosorbent assay (ELISA), phage ELISA, phage capture assay, electrochemical biosensor, total-prostate-specific antigen (t-PSA), free-prostate-specific antigen (F-PSA), electrochemical impedance spectroscopy, label-free immunosensor

## Abstract

Prostate cancer (PC) is the second most diagnosed cancer among men. It was observed that early diagnosis of disease is highly beneficial for the survival of cancer patients. Therefore, the extension and increasing quality of life of PC patients can be achieved by broadening the cancer screening programs that are aimed at the identification of cancer manifestation in patients at earlier stages, before they demonstrate well-understood signs of the disease. Therefore, there is an urgent need for standard, sensitive, robust, and commonly available screening and diagnosis tools for the identification of early signs of cancer pathologies. In this respect, the “Holy Grail” of cancer researchers and bioengineers for decades has been molecular sensing probes that would allow for the diagnosis, prognosis, and monitoring of cancer diseases via their interaction with cell-secreted and cell-associated PC biomarkers, e.g., PSA and PSMA, respectively. At present, most PSA tests are performed at centralized laboratories using high-throughput total PSA immune analyzers, which are suitable for dedicated laboratories and are not readily available for broad health screenings. Therefore, the current trend in the detection of PC is the development of portable biosensors for mobile laboratories and individual use. Phage display, since its conception by George Smith in 1985, has emerged as a premier tool in molecular biology with widespread application. This review describes the role of the molecular evolution and phage display paradigm in revolutionizing the methods for the early diagnosis and monitoring of PC.

## 1. Introduction

Prostate cancer (PC) is the second most diagnosed cancer among men [1]. It was observed that the early diagnosis of the disease is highly beneficial for the survival of cancer patients [2,3]. Therefore, the extension and increasing quality of life of PC patients can be achieved by broadening the cancer screening programs that are aimed at the identification of cancer manifestation in patients at earlier stages, before they demonstrate well-understood signs of the disease [2,3,4,5,6,7]. A significant social impact and economical effect of PC screening was estimated, taking into consideration that the cost for treating the advanced disease is much higher than the scanning cost [8]. Therefore, there is an urgent need for standard, sensitive, robust, and commonly available screening tools for the identification of early signs of cancer pathologies [9]. In this respect, the “Holy Grail” of cancer researchers and bioengineers for decades has been molecular sensing probes that would allow for the diagnosis, prognosis, and monitoring of cancer diseases via their interaction with tumor-associated cancer cells and/or blood-solubilized PC biomarkers, such as the prostate-specific antigen (PSA), the prostate cancer antigen 3 gene (PCA3), and sarcosine oxidase [2,10,11,12,13,14,15,16]. At present, most PSA tests are performed at centralized laboratories using high-throughput total PSA immune analyzers. There are clear practical advantages to using these fully automated analyzers, including lower detection limits and high-throughput samples. However, several authors noted a persistent disagreement among the PSA results obtained by different commercial immunoassays [17,18,19,20,21]. This serious drawback of the PSA immunoanalyzers can be attributed to the use of capture and tracer antibodies with different epitope specificities and affinities. Furthermore, an important limitation of PSA analyzers is that they are suitable for dedicated laboratories and are not readily available for the broad healthcare community. Therefore, the emerging trend in the screening and detection of PC is using portable biosensors for mobile laboratories and individual use [13,14,15,16,17,18,19,20,21,22]. Phage display, since its conception by George Smith, in 1985 has emerged as a premier tool in molecular biology with widespread application. This review describes the role of the molecular evolution and phage display paradigm in modernizing the methods for the early diagnosis and monitoring of PC.

## 2. Advanced Phage-Driven Analytical Tools for the Diagnosis of PC

New urgent requirements for fast, sensitive, accurate, and inexpensive tools for the early diagnosis of PC devalue the traditional PSA detection methods [16], such as ELISA, radioimmunoassay immunoradiometric assay, and time-resolved immunofluorescence assay, which have a complex operation, are difficult to miniaturize, and can have a limited sensitivity [23,24,25,26]. Modern immunoassays and biosensors require a biorecognition probe, which is attached to the interface of the analytical device, bind the target biological ligand, and participate in generating a measurable signal [13,14,27,28], as illustrated in Figure 1. For example, in electrochemical biosensors, the signal can be displayed in impedimetric, amperometric, or potentiometric formats.

To complement the p3-type phage display vectors, which were designed to discover therapeutic and diagnostic peptides and antibodies (Figure 2B) [29,30], the p8-type phage technology was developed with the goal creating diagnostic and detection nanoprobes by resurfacing the whole phage particles [31,32,33,34,35,36,37,38,39,40], as illustrated in Figure 2C and Figure 3. In the p8-type phage display system, called landscape phage, the dense array of foreign peptides on the body of the phage composes a unique organic landscape. The constrained conformation of individual peptides, influenced by interactions with neighboring proteins, can essentially increase their affinity to their counterpart ligands and receptors, in the same way that the scaffold of antibodies and other constrated phage-displayed molecules can determine the properties of fusion peptides [33,34,41,42,43,44], as reviewed in [45]. Therefore, each landscape phage particle can be treated as a unique nanomaterial with novel and emergent properties that cannot be observed by the use of an individual synthetic peptide alone [46]. In many applications, including the detection of soluble cancer biomarkers and cancer-cell-associated antigens, the unique architecture, extreme multivalency, and rigid scaffold of landscape phages are highly beneficial.

It was proved that the landscape libraries represent an inexhaustible, rich source of substitute antibodies—filaments that bind protein and glycoprotein antigens with nanomolar affinities and high specificity [25,33,35,36,37,44,48,49,50,51]. The foreign amino acids that form the biospecific ‘active site’ of a landscape phage comprise up to 25% of the total weight of the particle and up to 50% of its surface area, which can accommodate hundreds of bound protein antigens. More detailed information regarding the evolution of the landscape phage detector paradigm, starting from its appearance in 1996 as a distinct part of the phage display concept [42], which use the landscape phage as phage substitute antibodies in the first phage biosensor, and the development of the landscape-phage-based biosensors for the liquid biopsy of PC, can be found in the references [33,35,36,48,49,52,53,54,55,56,57,58,59,60,61,62,63,64,65].

The PC detection techniques can be divided into two major categories: (1) cancer cell imaging, and (2) cancer-cell-secreted soluble biomarker detection techniques [15]. Most analytic platforms rely on the use of monoclonal antibodies (mAb) as biorecognition probes. However, their broad application is limited by their high cost and intrinsic sensitivity to the components of body fluids [7,15,16,21,22,66,67,68,69,70,71,72]. Harnessing the power of molecular evolution, the phage display technique offers a new way of generating a rich repertoire of binding probes for any protein ligand or receptor. The idea of phage display as a molecular evolution tool lies in the genetic fusion of a foreign protein to the phage capsid and its preservation through the viral replication [29,30,31,42,73]. Considering the desirable characteristics of different display systems, filamentous bacteriophages M13 and fd were commonly preferred as suitable vectors for generating peptide- and antibody-fusion phage-displayed libraries.

### 2.1. Selection of Phage Probes against PC-Cell-Associated Antigens

#### 2.1.1. Selection of PC-Cell-Binding Phages from f8-Type (Landscape) Libraries

Since it was proven that the malignant transformation of cells is linked with the expression of cell antigens, the tumor-cell-specific phage-displayed peptides and antibodies were considered prospective, versatile diagnostic and therapeutic reagents [40,74]. The first PC-cell-targeting landscape phages were discovered by Victor Romanov and colleagues [52]. It was shown that the phage-displaying N-terminal 8 mer peptide DPRATPGS was inserted in all 4000 domains of major coat protein p8, selected from the p8-type (landscape) library f8/8 (Figure 4) according to its affinity selection (biopanning), against PC cells LNCaP and their relatives C4-2 and C4-2b (Figure 5) [32]. Later, the major principles and methods of phage selection were used in the publications of other groups, but some details of the protocol were modified with the purpose of increasing the specificity and selectivity of the discovered phage probes towards the target cancer cells.

To continue this pioneering work, Prashanth Jayanna et al. used PC3 cells as target cells, as they imitate the profile of advanced prostate tumors [75]. To increase the repertoire of binding phages, the f8/9 (9-mer) library was used, in addition to the f8/8 (8-mer) library (Figure 4) [38,43]. To isolate phages with high selectivity towards PC-specific antigens, the libraies were depleted against plastic, the serum, and normal fibroblast cells before being allowed to interact with the target cells. elative affinity of selected clones towards targeted and control cells was estimated using a selectivity assay, which is based on the interaction of phage particles with PC-3M cells in comparison with other control cells (Figure 6). The affinity of phage DTDSHVNL to PC3 cells was ~9 times higher than that to either of the control cells, and 32 times higher than that to serum (cell-free media), whereas the affinity of phage DTPYLDTG to PC3 cells was ~8 times higher than to either of the control cells and 15 times higher than to serum. Note: Landscape phages were designated by the sequences of the inserted 8-mer and 9-mer peptides. The other clones that were analyzed showed a high affinity with the target cells, as well as the control cancer cells, but not to normal epithelial cells or serum, leading us to assume that these probes may be directed against a universal cancer receptor. A single clone, DVVYALSDD, isolated from the 9-mer library demonstrated an affinity to PC3 cells that was almost 80 times higher than its affinity to the control cells and 600 times than its affinity to serum (cell-free media). Surprisingly, the other analyzed clones showed high affinity to the target as well as normal epithelial cells, but not to control cancer cells or serum, indicating that they may be directed towards a receptor that is common to both tumor and normal cells. A phage bearing an unrelated streptavidin–avid peptide (VPEGAFSS) was used a control to demonstrate the specificity of the landscape phage probes.

To extend the panel of PC-imaging phage probes, Olusegun Fagbohun et al. screened landscape phage library f8/8 against metastatic PC cells PC-3M [53]. The most selective for PC-3M cells, phage EPTHSWAT, was able to penetrate the PC-3M cells, as revealed by immunofluorescence microscopy (Figure 7). The selectivity of the PC-specific phages EPTHSWAT towards PC-3M cells was studied by the phage-capture assay and demonstrated 35-fold greater binding than the non-relevant control phage VPEGAFSS. Furthermore, phage EPTHSWAT showed a statistically significant higher interaction with PC-3M cells than with other cells, RWPE-1, HT-29, and serum. This high interaction of the phage particles with PC-3M cells might be due to a specific phage’s interaction with an overexpressed PC cell antigen.

#### 2.1.2. Selection of PC Cell Binders from p3-Type Phage-Displayed Antibody Libraries

The polyclonal antibodies purified from the serum of an immunized animal (mouse, rabbit, goat, lama, etc.) and mAb secreted by immortalized B cells from the spleen of an immunized animal are commonly used in immunological assays. Their dominant role in immunochemical applications faded after the appearance of phage-displayed antibodies, which are currently commonly used for the discovery and detection of cancer-specific antigens and biomarkers [28,30,76,77]. To isolate antibodies with the desired specificities, phage library selections must be performed on tumor-derived antigen sources. For example, the phage display strategy for the selection of rabbit monoclonal antibodies that recognize PC tumor-associated antigens was reported by Mikhail Popkov et al. [77]. Researchers immunized rabbits with either human PC cell line LNCap or DU145. Chimeric rabbit/human Fab libraries were generated through the oligo(dT)-primed, reverse transcription of RNA from the animal’s spleen and bone marrow [78]. The antibody variable domains VL and VH were amplified, fused to human constant domains CL and CH1, and cloned into the phagemid vector pComb3X. Constructed in this way, phage-displayed chimeric rabbit/human Fab libraries were screened against human PC cells DU145 using a novel whole-cell panning protocol, resulting in the discovery of clones bound selectively to DU145 cells but not to primary human prostate epithelial cell line PrEC, as detected by flow cytometry. In summary, this work first demonstrated the potential of immune antibody libraries in the identification of imaging phage probes interacting with tumor-associated cell-surface antigens. A comprehensive review describing the progress in the preparation and use of recombinant phage-displayed antibodies for bioanalytical applications was published recently by Guliy et al. [30].

### 2.2. Selection of Phage Probes against Prostate-Specific Antigen (PSA)

#### 2.2.1. PSA as a PC Biomarker

PSA is a serum marker that is commonly used for the diagnosis of prostatic diseases. Normally, it is produced by epithelial cells of the prostate and mainly exists in two molecular forms: free PSA (f-PSA), 10–30% of PSAs, and the PSA-α1-antichymotrypsin complex (PSA-ACT), 70–90% of PSAs [79,80,81,82]. The sum of f-PSA and PSA-ACT is called the total PSA (t-PSA), which is regarded in clinical medicine as the important index for the early diagnosis of PC, an evaluation of the curative effect, and the post-operation monitoring [7,83,84,85,86,87]. In general, the content of t-PSA in the serum of healthy people is lower than 4 ng/mL, the level accepted as a threshold value in the clinical test of PC. When the content of t-PSA in serum is more than 10 ng/mL, the risk of PC is high; thus, the accuracy rates of the diagnosis of PC can reach 70–80% [88,89,90,91]. Therefore, joint detection of the ratio of f-PSA/t-PSA and the level of t-PSA can more accurately discriminate PC and prostate diseases. However, there is skepticism regarding the diagnostic and prognostic significance of the fPS/tPSA ratio because of the non-standard nAb used in the PSA diagnostic tests [83,92,93].

#### 2.2.2. Selection of p3-Type Phage-Displayed Peptides against PSA

To obtain peptide ligands that specifically recognize different forms of PSA, phage-displayed linear and cyclic peptide libraries were screened against PSA-coated microplate wells or PSA supported by immobilized anti-total PSA mAbs [94,95]. In their pioneering work, Ping Wu et al. [96,97,98] discovered PSA-binding peptides by screening p3-type cyclic and linear peptide phage-display libraries. A p3-fused cyclic peptide with four bridged cysteine residues showed the highest affinity for PSA. The binding specificity was characterized by competition with the monoclonal anti-PSA antibodies of known epitope specificities. The peptides bound to the same region as mAbs specific for free PSA, indicating that they bind close to the active site of the enzyme. These results demonstrated that peptides binding to PSA and modulating its enzyme activity can be developed by the phage-display technique. However, when the discovered peptides were tested in sandwich capturing PSA assays with the anti-PSA 5D5A5 mAb, the lowest concentrations of detectable PSA was 0.2–2 mg/mL, which was not sensitive enough to allow PSA to be quantified in sera from patients with prostatic diseases where the concentration of PSA is greater than 2–4 ng/mL. In the recent study of Wang et al. [95], the elution strategy in the biopanning of p3-type phage-displayed peptide library Ph.D.–12 (New England Biolabs, Ipswich, MA, USA) against PSA was optimized by additional BSA pre-screening and serum interference. PSA-specific phage-expressing peptide TSIANYIGLALR showed the best affinity and specificity against PSA and was conjugated through C-terminal GGGGSK-biotin linker to streptavidin. This construct was used as a signal amplifier in the sandwiched ELISA system. The assay could detect total prostate-specific antigen (tPSA) with a linear range of 0.25–200 ng/mL and detection limit of 0.18 ng/mL, demonstrating the good prospects of using peptide–streptavidin conjugates as substitute signaling antibodies in t-PSA.

#### 2.2.3. Development and Affinity Maturation of p3-Type Phage-Displayed Antibodies against PSA

The power of directed evolution and phage display was successfully used by Muller et al. to enhance the affinity and sensitivity of the immunoassay while maintaining its selectivity [99]. The original f-PSA assay, based on the use of the high-off-rate 4D4 Mab as a tracer, was less than ideal regarding the sensitivity and low-end robustness of the assay. Attempts to use the 4D4 Mab for capturing were also not successful. Using phage-display library-derived mutant L3-2 Fab with reduced off-rate dissociation constant, both configurations were possible and improved assay performances. In comparison with the wild-type scFv, the best binders showed an enhancement of sensitivity in sandwich immunoassay.

#### 2.2.4. Selection of PSA-Binding p8-Type Multivalent Landscape Phage Probes

Target-specific landscape-phage probes can be prepared as described in commonly available protocols [25,37], as illustrated in (Figure 8). Thus, the specific phage probes against f-PSA and t-PSA were selected from the f8/8 landscape phage library [49].

Through three rounds of biopanning and phage-capture assay, novel phage clone P1- and P5-displaying octamers ERNSVSPS and ATRSANGM with the best affinity and selectivity for t-PSA were identified and used as the capture probes to establish both ELISA and the electrochemical impedance spectroscopy (EIS) assay systems (Figure 9, Figure 10, Figure 11 and Figure 12).

## 3. Development of Phage-Driven Biosensors for the Detection of Different forms of PSA

The critical factor that determines the efficacy of early cancer detection is the analytical platform that converts invisible molecular binding events into optical or electrical signals (Figure 1). The common methods used for the detection of cancer-specific antigens (biomarkers) include ELISA, and different types of immunosensor devices [16,48,101,102]. To date, most analytical platforms for the detection of cancer biomarkers use mAb as a sensing probe [10,14,20,68,70,103,104,105,106,107,108,109,110]. There is an urgent need for robust, inexpensive, highly sensitive, and easily available sensing probes, such as landscape phage substitute antibodies [33,42]. It was shown that the presentation of ~400 copies of heavily constrained diagnostic peptides on the surfce of the landscape phage particles allows for a a dense interface to be formed on the detection platorms with significantly increased affinity and selectivity towards the analyte markers in comparison with individual randomly conjugated peptide binders with a mostly disordered conformation [35,50,111,112].

### 3.1. Landscape-Phage-Driven Enzyme-Linked Immunosorbent Assay (Phage ELISA)

There are multiple variants of the ELISA technique [113,114], which include the sandwich ELISA, in which the immobilized capture antibody binds the water-soluble target antigen to form a complex, which is detected by a detector antibody that binds the captured antigen and produces a visible signal after adding a chromogenic substrate, as illustrated in Figure 1a. The landscape phage can be used in sandwich ELISA as a substitute for both capture and detection antibodies, as reviewed in [100,115]. As a substitute antibody, the landscape phage leverages the uniqueness of the p8 display system described above (Figure 1b). The landscape-phage-based signaling antibody also can benefit from phage multivalency. The t-PSA- and f-PSA-specific octapeptide-fusion landscape phages were selected from the f8/8 landscape-phage-displayed library (Figure 7), as described in the legend to Figure 8, Section 2.2.4, and related publications [25,61]. Phages selected against immobilized recombinant t-PSA and f-PSA showed the best affinity and selectivity as capture probes in a model ELISA (Figure 10) and demonstrated good sensitivity and reliability in the t-PSA and f-PSA analysis in real serum samples. This work first proved that a phage-based immunoassay can be used for the optimization of phage structure during the development of advanced PSA detection systems.

### 3.2. Phage-Driven Electrochemical Immunosensors for Detection of PSA

Electrochemical immunosensors attracted the attention of bioengineers as bioanalytical platforms for PC detection because of their high sensitivity, specificity, simple operation and easy miniaturization. Among the numerous electrochemical methods, the electrochemical impedance spectroscopy (EIS) not only inherits the normal advantages of electrochemical immunosensor, but also shows ultra-high sensitivity [70,116,117,118,119,120,121]. Like other immune assays, EIS technology is based on the specific immunological recognition of the ligand with the antigen [122,123]. Considering the unique smart nanomaterial properties of landscape phages, Lei Han et al. constructed the first phage-based dual f-PSA/t-PSA ratio assay [49]. As sensing probes for the construction of the sensor’s interface the researchers used landscape phages that were discovered through screening the f8/8 landscape phage library against f-PSA and t-PSA [25]. The fabricated immunosensors showed high specificity and selectivity, an ultra-low limit of detection, a wide linear detection range, excellent reusability, high reproducibility, and good stability, corresponding to the high stability and biocompatibility of the fibrous phage interface. Specifically, the phage-driven sensors demonstrated wide linear ranges (0.02–200 ng mL^−1^) for f-PSA; 0.02–200 ng mL^−1^ for t-PSA (Figure 11 and Figure 12).

The encouraging results of this study provided a novel avenue for the construction of phage-based sensors for the dual detection of f-PSA and t-PSA and analysis of the f-PSA–t-PSA ratio in human blood (Table 1). One can assume that landscape phage substitute antibodies against different forms and multiple epitopes of PSA can be obtained by using a more diverse landscape phage library f8/9 for biopanning [43] and developing novel more advanced phage-based immunosensors, as reviewed in [16].

To evaluate the practical significance of landscape-phage-based dual immunosensors, they were used to determine t-PSA and f-PSA/t-PSA in clinical serum samples, as shown in Table 1, adapted from [48].

Each sample was determined for three independent detections using different freshly constructed immunosensors. As shown in Table 1, the obtained results were in agreement with the established values and the relative errors were within 3%, indicating that the systems could reliably and reproducibly detect the concentrations of f-PSA and t-PSA in real serum samples.

To summarize, the fabrication of specific phage-based immunosensors for the ultrasensitive detection of f-PSA and t-PSA in human sera is an encouraging example of effectively harnessing the power of molecular evolution and phage display for the creation of smart detection materials that can recognize and monitor the presence of different forms of prostate cancer biomarkers in human blood [16].

## 4. Conclusions

The analysis of the literature data shows the good prospects of using phage-display methods for the design of analytical tools for the early detection of cancer, specifically prostate cancer. Traditionally, phage-display methods have been used for the discovery and improvement of antibodies, and their antigen-binding fragments are widely used in numerous immunochemical detection devices. Considering the huge public interest in this area of research and development [76], one can expect the further development of antibody phage display technology to improve the performance of modern detection methods. Furthermore, the tremendous success in the development of novel analytical methods for the fast, inexpensive detection of SARS viruses using novel analytical platforms and sensing materials during the COVID pandemic promoted the development of new approaches for the construction of novel biosensor platforms for the mass screening of humans for early signs of cancer. In this contest, the landscape phages described here appear competitive in comparison with novel interfaces and diagnostic probes [16]. For example, the model-phage-based electrochemical and ELISA immunosensors showed comparable limits of detection to most advanced antibody-, aptamer-, and Au-based biosensors. In contrast to other interfaces, the phage structure is extraordinarily robust, being resistant to heat, organic solvents, urea, acid, and alkali, and can tolerate different modifications that increase the efficacy of their use as detection and diagnostic probes [124]. For example, the conjugation of a phage with gold nanoparticles is a straightforward, reliable process [62], which can be easily adapted for the construction of different types of lateral flow immunoassays for point-of-care prostate-specific antigen testing [16]. Purified phages can be stored indefinitely at moderate temperatures without losing their infectivity and binding activity [46,125,126]. An important advantageous characteristic of landscape phages is their availability for safe, large-scale production, and their use in prostate cancer control, as recently reviewed [16,127,128,129].

## Figures and Tables

**Figure 1 viruses-16-00277-f001:**
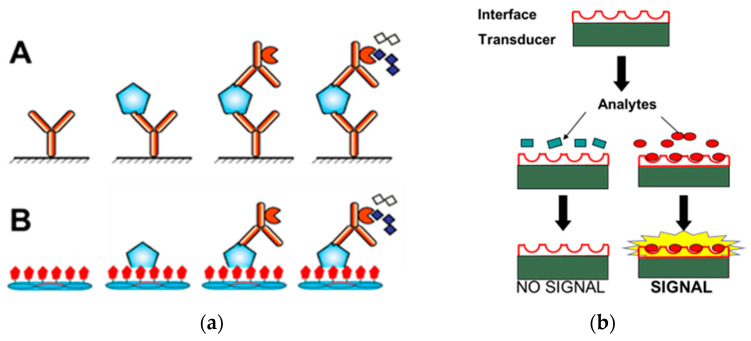
(**a**): Sandwich PSA ELISA (**A**) vs. Phage PSA ELISA (**B**): The capture antibody (**A**), or phage (**B**) immobilized onto ELISA plates bind the analyte protein, and the detection antibodies linked to the enzyme are added to catalyze the appearance of a colored or fluorescent product. (**b**): PSA biosensors. A molecular interface linked to a transducer binds the analyte and generates a signal for changes in mass, capacitance, resistance, surface plasmon resonance, etc.

**Figure 2 viruses-16-00277-f002:**
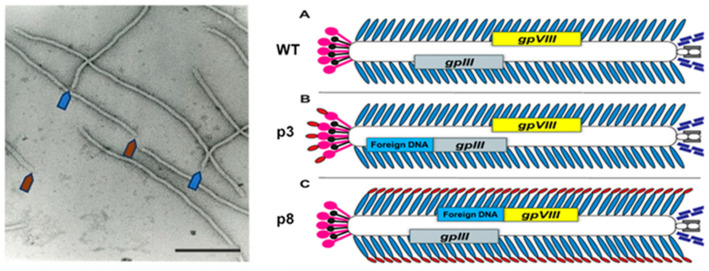
(**Left**) Electron microscopy image of the wild-type phage fd. Blue and red arrows depict the sharp and blunt ends of the phage capsid with the attached minor coat proteins p3/p6 and p7/p9, respectively (five copies each). Major coat protein p8 (~2700 copies) forms the tubular capsid around viral single-stranded DNA (scale bar: 100 nm, the length of the phage capsid ~1 µm). (**Right**) Peptide phage-displayed libraries. There are two essential types of phage display—display in the minor coat protein p3, and display in the major coat protein p8. (**A**) Phage vector fd-tet composed of 4000 copies of the p8 major coat protein (blue) and five copies of minor coat proteins p3 (pink), pVI (black), pIX (gray) and pVII (purple) each. (**B**,**C**) p3 and p8 phage display libraries. A random peptide (red) is fused to every copy of either p3 or p8 proteins. Adapted with modifications from [47]. The nomenclature of the types of phage display systems is provided in the phage display review [42].

**Figure 3 viruses-16-00277-f003:**
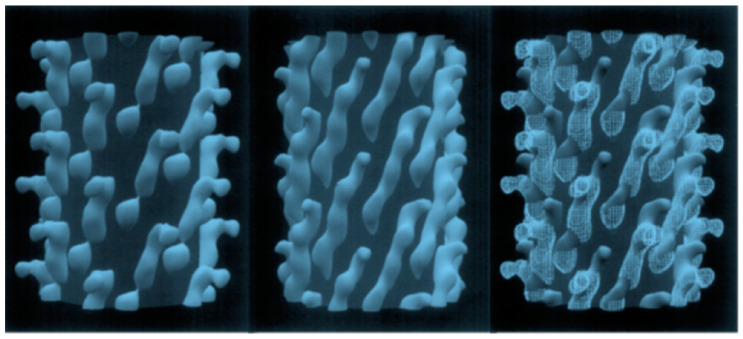
Configuration of a 5-mer peptide displayed on bacteriophage (phage) M13. Computer rendering of a ~10 nm length for the surface of electron density maps of M13 (**left**), fusion phage with 5-mer peptide inserted in all copies of p8 proteins (**center**), and a rendering of the differences between images (**right**). A cylinder of 2.5 nm radius was added to images to mask the essentially identical interior features of the phages. About half of each coat protein is visible in phage surface images. Adapted from [46].

**Figure 4 viruses-16-00277-f004:**
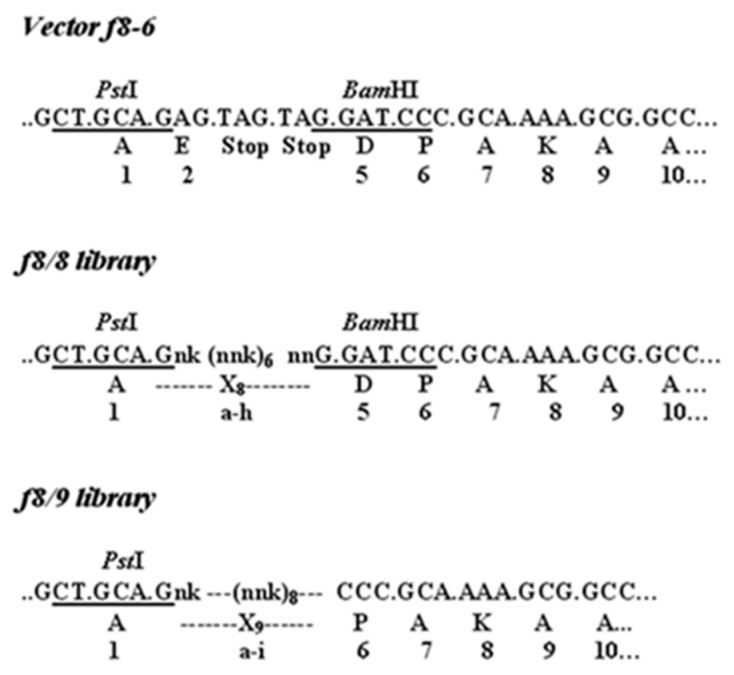
Vectors and libraries. In the nucleotide sequences corresponding to the part of recombinant gene *gpVIII*, encoding the N-terminal part of the major coat protein, randomized structures are designated as nnk, where n = A, T, G, or C, and k = G or T. Restriction sites for PstI and BamHI are underlined. N-terminal amino acid structures of mature recombinant pVIII proteins in libraries are indicated by capital single letters according to amino acid abbreviations. Randomized amino acids are designated by small letters (a–h in the f8/8 library and a–i in the f8/9 library). Amino acids are numbered as in vector phage f8-6 [41]. The nomenclature of the types of phage display vectors and libraries is provided in [38,42].

**Figure 5 viruses-16-00277-f005:**
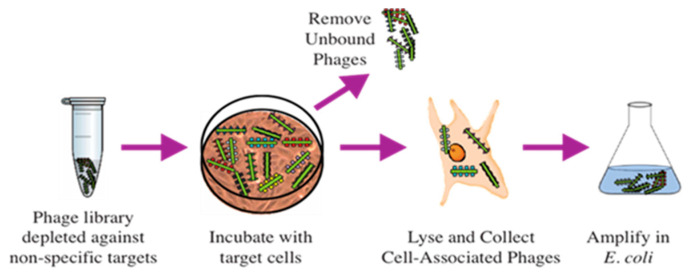
Selection of landscape phage interacting with PC-cell-associated antigens. The most common phage survey strategy is affinity selection, called ‘biopanning’ which enriches phage particles whose displayed peptides bind the target cells in culture or whole tissues in living animals. To use biopanning for selection of landscape phages against a variety of different PC, the researchers add the library to the immobilized target cells, wash away non-bound phage, elute bound phage particles, and amplify them. After 2–4 rounds of selection, they propagate individual clones, and analyze them. This procedure was named biopanning, because it is reminiscent of panning—the process of extraction of gold particles from sand.

**Figure 6 viruses-16-00277-f006:**
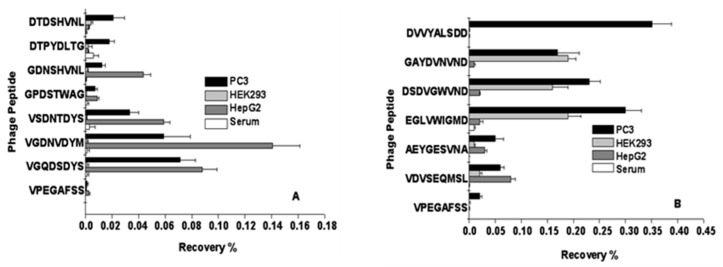
Selectivity and specificity of phage probes. Phage probes selected from preliminary screening assays were incubated with target PC3 cells, control cells, or serum-treated wells of a 96-well cell culture plate. Phages associated with cells or serum were titered in bacteria and the ratio of phage output to phage input was expressed as recovery % to obtain the measure of the selectivity of a particular clone. The % recovery of the control phage bearing an unrelated peptide relative to the selected phage probe was indicative of the probe’s specificity. (**A**) f8/8 library, (**B**) f8/9 library. Adapted from [56,75] with permission from Elsevier.

**Figure 7 viruses-16-00277-f007:**
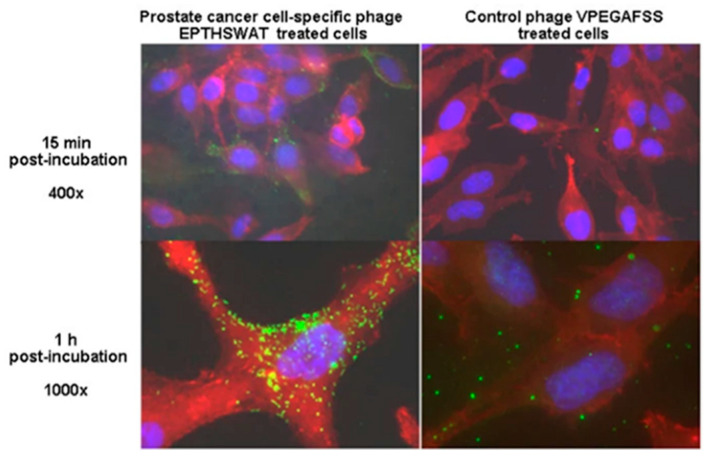
Immunofluorescence microscopic demonstration of phage EPTHSWAT’s interaction with PC-3M cells at 15 min and 1 h in comparison with the control non-relevant phage VPEGAFSS. Adapted with modifications from [53].

**Figure 8 viruses-16-00277-f008:**
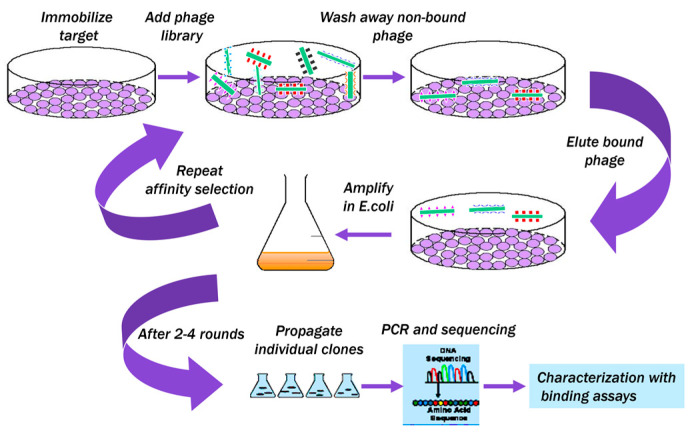
Schematic illustration of biopanning for t-PSA (f-PSA and PSA-ACT). The f8/8 landscape phage library was added to the dishes with different immobilized forms of PSA. Unbound phages were washed away, and bound phages were eluted and used as a sub-library in the next round of biopanning. After three rounds, the individual phage clones were propagated, and their DNA segments corresponding to gpVIII were sequenced to determine the corresponding phage-displayed peptide sequences. Detailed procedures can be found in [25,37,100].

**Figure 9 viruses-16-00277-f009:**
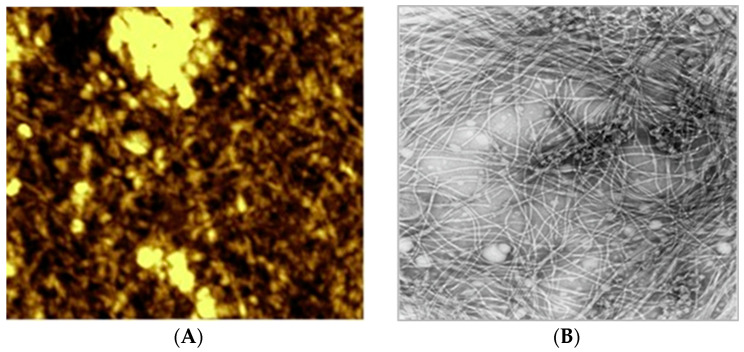
The phages can be conjugated to the gold electrode surface using carbodiimide chemistry (**A**), or immobilized to the gold sensor through physical adsorption [36] (**B**) and analyzed by atomic force microscope (**A**) or electron microscopy (**B**). As shown in panels (**A**,**B**), myriads of filamentous phage particles were attached to the gold surface to generate an intercrossing random network. Adapted with modifications from [36,49] with permission from Elsevier.

**Figure 10 viruses-16-00277-f010:**
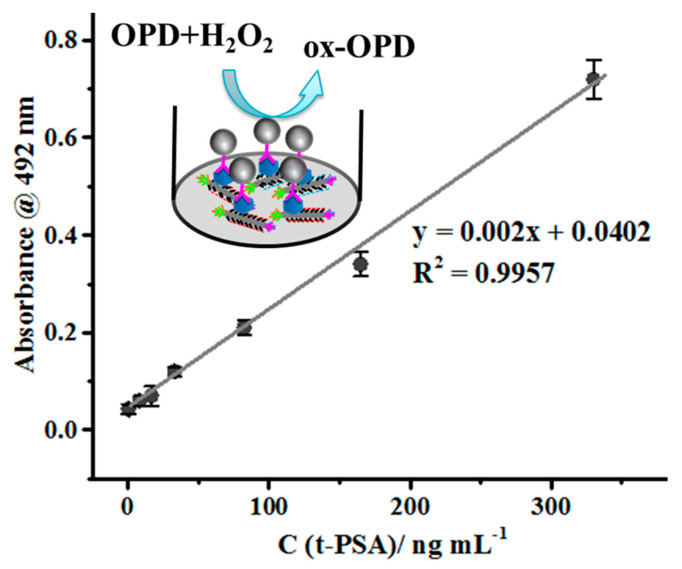
Standard curve for t-PSA. The schematic illustration of the phage-based ELISA for t-PSA detection is shown in the insert. The selected phage was loaded into wells of a 96-well plate. After coating the wells overnight, probes with different concentrations of t-PSA were added, and the plate was incubated at 37 °C. Then, mAb, IgG-HRP, and OPD were added successively. The absorbance values at 492 nm were linear with t-PSA concentration from 3.3 to 330 ng mL^−1^. The limit of detection was calculated to be 1.6 ng mL^−1^. Adapted from with permission of ELSEVIER [25,49].

**Figure 11 viruses-16-00277-f011:**
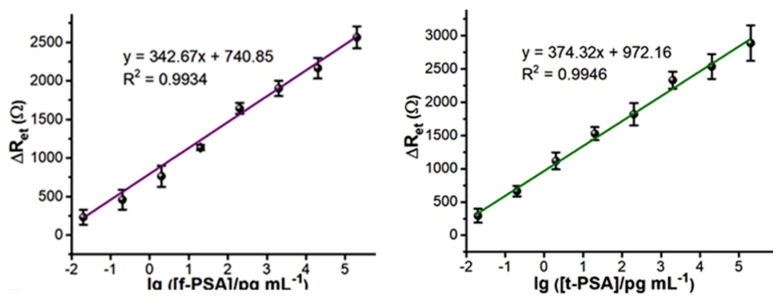
The calibration curves of dual immunosensors after immunological recognition with different concentrations of f-PSA (**left**) and t-PSA (**right**). Adapted from [49] with permission of Elsevier.

**Figure 12 viruses-16-00277-f012:**
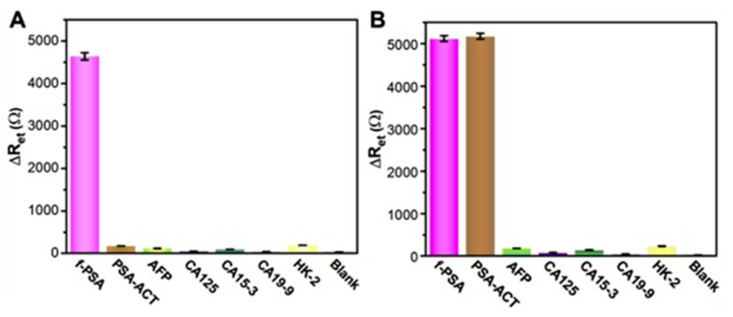
The selectivity assay of P1- ERNSVSPS immunosensor (**A**) and P5- ATRSANGM immunosensor (**B**) for f-PSA and t-PSA (total of f-PSA, and PSA-ACT) in comparison with other common cancer biomarkers (AFP, CA125, CA15-3, CA19-9, and hK-2) present in the serum as controls. The samples were dropped onto the phage-covered immunosensors and incubated for same time. EIS assay was performed for the above immunosensors as described. Adapted from [49] with permission from Elsevier.

**Table 1 viruses-16-00277-t001:** Analysis of f-PSA–t-PSA ratio in human blood.

	f-PSA (ng mL^−1^)			t-PSA (ng mL^−1^)			f-PSA/ t-PSA Ratio
	Known	Detected	Relative Error (%)	Known	Detected	Relative Error (%)	
1#	1.26	1.24 ± 0.04	−1.58	4.92	4.81 ± 0.13	2.23	0.258
2#	1.71	1.76 ± 0.05	2.92	6.83	6.98 ± 0.21	2.19	0.252
3#	3.58	3.67 ± 0.10	2.51	10.21	10.47 ± 0.35	2.54	0.351

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
