# Peer review of "Phage Display’s Prospects for Early Diagnosis of Prostate Cancer"

_viruses, 2024, doi:10.3390/v16020277_

Round 1
Reviewer 1 Report
Comments and Suggestions for Authors
The paper (Introduction) recalls the clinical relevance and challenges of prostate cancer (PC) diagnosis and follow-up evaluations.
In the following sections, the paper provides an excellent overview of the very large field of developments of phage display – biopanning to identify biomarkers and cancer specific antigens by focusing on prostate PC research.
Indeed, PC diagnosis is a very good example to recall the spectrum of different aspects of the various types of phage constructs and their individual “construct” advantages in optimized target recognition and in further secondary signal amplification by divers applied modifications (immunochemistry, chemical labeling etc).
The schematic illustrations provide appropriate support for an easier understanding and recalling the differences in technical approaches.
Important is furthermore the recall of the comparative evaluation of various PC diagnosis assays by providing /recalling the corresponding data in Figures.
Interesting and important is the personal view of the author when insisting in phage displayed advantages for diagnosis in comparison to the commonly used antibody assays by most screening / clinical testing platforms.
The reference list presents many original investigations of the author of the present review paper. This is justified by the very large contribution of the author and the coworkers to the field for several decades.
The text needs revision for minor editorial / typographical errors like:
Line 221 di – seases
Line 227 in the , sandwich ELISA
Line 310 [62] . .
There might be others.
Author Response
Thank you very much for your comments, suggestions, and critic.
All observed defects in the text have been corrected.

Reviewer 2 Report
Comments and Suggestions for Authors
The manuscript nicely reviews the uses of phage display in prostate-cancer-related research, thus potentially benefiting future studies and patients. Yet, in addition to some minor suggestions, I would highly recommend (a) further discussing t-PSA vs. f-PSA from the perspectives of detecting mechanisms and diagnosing implications and (b) clarifying the advantages of multi-valent landscape phages. See more details below.
(1) It is suggested to provide full names of the acronyms upon their first appearance, such as PSA and PCA3 in line 41 as well as DPV in line 249.
(2) Kindly proofread the manuscript for minor formality issues and clerical errors, such as “cellassociated” in line 14, “correpondinly” in line 15, missing a “.” in lines 29, 48, 141, and 288, “andother” in line 41, duplicate “,” in line 48, duplicate “.” in lines 79, 99, 127, 284, and 310, “sselection” in line 114, “zdapted” in line 159, “assayand” in line 164, “pspecific hage” in line 167, “generatedthrough” in line 183, “It” and “exist” in line 195, “di-seases” in line 221, “by by” in line 224, “the, sandwiched” in line 227, “ignal” in line 312, “.:” in line 330, a missing bracket in line 349, and “a encouraging” in line 367. Further, a “which” is suggested to be added in line 198 before “is.”
(3) Concerning the sentences starting in lines 72 and 73, does the latter one flow better from the previous paragraph? Accordingly, does it make sense to delete or move the former one to another place, such as right before Section 2.1?
(4) Concerning Figure 2, what do the green circles above PSA represent? If the signaling substrate is with PSA, do we still need the top layer of antibodies as illustrated in Figures 1aA and 1aB? Or, does the substrate represent the substrate of PSA? If so, kindly clarify it in the manuscript to avoid any confusion.
(5) It seems the scale bar noted in line 97 is missing from Figure 3, left panel.
(6) For the ease of our future readers’ understanding and potentially better flow, does it make sense to change “f8-type (landscape)” in line 105 to “p8-type (landscape)” as it is introduced in line 111?
(7) Shall we update “Figure 3, 4” in line 112 to Figures 4 and 5?
(8) In Figure 4, the parent vector is labeled as f8-6 in the figure, but noted as f8-5 in the legend (see line 124). Shall we use one for consistency? Or, do they refer to different vectors?
(9) Shall we add a “having” between lines 137 and 138?
(10) Shall we delete “is” in line 140?
(11) Kindly elaborate in the manuscript (preferably following line 86) on why the p3 libraries were used to screen for the antibodies while the p8 libraries were used for peptides. Further, kindly comment on the achievements of the p3 screening illustrated in Section 2.1.2.
(12) Concerning the sentence starting in line 200, I am not sure the reasons why different PSA sub-forms would “permit discrimination between benign and malignant cases” have been provided. Similar comments apply to the sentence starting in line 204 and the title of Section 3.
(13) From lines 243-250, I can see the importance of the multi-valency of landscape phages if we would like to distinguish a cell expressing multiple copies of PSA from free PSA. Yet, I cannot imagine why multi-valency will be important when detecting free PSA or total PSA using peptides displayed by phages. Would the author mind elaborating a little more here? Along this line, does the “bispecific” in line 253 refer to monospecific but bi-valent/multi-valent? Further, the advantages of multi-valency and stability of a landscape p8 phage display detailed in the paragraph starting in line 243 seem to fit better in the earlier section upon introducing the technology. Since its uses have been discussed earlier, for example, in Section 2.1.1, it is suggested to focus on using it in screening for a PSA probe in Section 2.2.4 as suggested in its section title in line 242.
(14) Does it make sense to remove the reference to Figure 10 in line 280?
(15) Does it make sense to update “Fig. 1B” in line 298 to Figure 10B?
(16) Concerning the sentence starting in line 303, differences between direct and indirect ELISA are commonly noticed as requiring a secondary antibody to amply the signal or not, instead based on the molecule to be detected. See, for example, https://www.bio-rad-antibodies.com/elisa-types-direct-indirect-sandwich-competition-elisa-formats.html. Kindly amend this sentence to avoid any confusion.
(17) I am not sure what the bold font in Section 3.1 refers to.
(18) Concerning the sentence in line 312 and further to the point (13) as made above, I am not sure multi-valency will help to amplify the signal. I understand multi-valency means one binder (i.e., the multivalent phage) binds to multiple copies of the target (e.g., PSA), and further, the signal-generating moiety should be with the binder. Accordingly, multiple copies of the target binding to one binder will activate only one signal-generation moiety. Will it actually considered the opposite of signal amplification? Or, do we intend to use the phage as the capturing moiety and use another labeled antibody to generate the signal? If this is the case, will a nanoparticle carrying or a plate coated with the same peptide as the phage work similarly? Also, if this is the case, is it the reason why this method can distinguish free PSA since PSA expressed on cells after binding to a phage most likely hindered from binding to another antibody? In other words, I would really appreciate some summary on why, after the binder screening, the phage is still critical to have as a part of the detection method. Along this line, some examples of phage detection methods approved by a regulatory authority, such as FDA, would be really helpful, even if they may not relate to PC.
(19) Further to the sentence in line 315, I would really appreciate some discussions on why and how phage can distinguish immobilized f-PSA from immobilized but not free PSA. Are the binding sites in the PSA different?
(20) After reading the current manuscript, I feel it seems premature to claim the use of phage displays for early diagnosis of prostate cancer. For example, after arriving at a peptide, an antibody, a Fab, or a scFv from a phage display screening, is it still advantageous to use them when they are displayed on phages for diagnosis? If the multi-valence of a landscape phage is critical, will coated gold nanoparticles serve as a similar (if not better) detection tool, such as the commercially available COVID test kits? For those electrochemical immunosensors, will they still require expensive equipment and centralized laboratories? In addition, it seems the current draft does not provide any information linking PSA, t-PSA, f-PSA, or any of their ratios to PC patients in their early stages. Further, I do not see any discussion on the specificity and selectivity data of the selected peptides using patient samples, thus, it may be a little early to claim any clinical indications.
Author Response
Responses to Reviewer 2.
Thank you very much for your comments, suggestions and critic that helped me to prepare the revised version of my manuscript. I believe that the new improved version of the review would fit better to the scope of the VIRUSES and allow understanding of prostate cancer detection problems that can be solved by using phage display technique.
Please find below the responses to your comments (capital)
Comments and Suggestions for Authors
The manuscript nicely reviews the uses of phage display in prostate-cancer-related research, thus potentially benefiting future studies and patients. Yet, in addition to some minor suggestions, I would highly recommend (a) further discussing t-PSA vs. f-PSA from the perspectives of detecting mechanisms and diagnosing implications and (b) clarifying the advantages of multi-valent landscape phages. See more details below.
- It is suggested to provide full names of the acronyms upon their first appearance, such as PSA and PCA3 in line 41 as well as DPV in line 249.
REVISED
- Kindly proofread the manuscript for minor formality issues and clerical errors, such as “cellassociated” in line 14, “correpondinly” in line 15, missing a “.” in lines 29, 48, 141, and 288, “andother” in line 41, duplicate “,” in line 48, duplicate “.” in lines 79, 99, 127, 284, and 310, “sselection” in line 114, “zdapted” in line 159, “assayand” in line 164, “pspecific hage” in line 167, “generatedthrough” in line 183, “It” and “exist” in line 195, “di-seases” in line 221, “by by” in line 224, “the, sandwiched” in line 227, “ignal” in line 312, “.:” in line 330, a missing bracket in line 349, and “a encouraging” in line 367. Further, a “which” is suggested to be added in line 198 before “is.”
CORRECTED
- Concerning the sentences starting in lines 72 and 73, does the latter one flow better from the previous paragraph? Accordingly, does it make sense to delete or move the former one to another place, such as right before Section 2.1?
CORRECTED
- Concerning Figure 2, what do the green circles above PSA represent? If the signaling substrate is with PSA, do we still need the top layer of antibodies as illustrated in Figures 1aA and 1aB? Or, does the substrate represent the substrate of PSA? If so, kindly clarify it in the manuscript to avoid any confusion.
REVISED.
- It seems the scale bar noted in line 97 is missing from Figure 3, left panel.
FIXED
- For the ease of our future readers’ understanding and potentially better flow, does it make sense to change “f8-type (landscape)” in line 105 to “p8-type (landscape)” as it is introduced in line 111?
The Figures 3 and 4 are supplied with the reference to the Phage Display review [40]..
- Shall we update “Figure 3, 4” in line 112 to Figures 4 and 5?
DONE
- In Figure 4, the parent vector is labeled as f8-6 in the figure, but noted as f8-5 in the legend (see line 124). Shall we use one for consistency? Or, do they refer to different vectors?
CORRECTED
- Shall we add a “having” between lines 137 and 138?
DONE
- Shall we delete “is” in line 140?
DONE
- Kindly elaborate in the manuscript (preferably following line 86) on why the p3 libraries were used to screen for the antibodies while the p8 libraries were used for peptides. Further, kindly comment on the achievements of the p3 screening illustrated in Section 2.1.2.
REVISED AS RECOMENDED
- Concerning the sentence starting in line 200, I am not sure the reasons why different PSA sub-forms would “permit discrimination between benign and malignant cases” have been provided. Similar comments apply to the sentence starting in line 204 and the title of Section 3.
THE CLAIM REMOVED
- From lines 243-250, I can see the importance of the multi-valency of landscape phages if we would like to distinguish a cell expressing multiple copies of PSA from free PSA. Yet, I cannot imagine why multi-valency will be important when detecting free PSA or total PSA using peptides displayed by phages. Would the author mind elaborating a little more here? Along this line, does the “bispecific” in line 253 refer to monospecific but bi-valent/multi-valent? Further, the advantages of multi-valency and stability of a landscape p8 phage display detailed in the paragraph starting in line 243 seem to fit better in the earlier section upon introducing the technology. Since its uses have been discussed earlier, for example, in Section 2.1.1, it is suggested to focus on using it in screening for a PSA probe in Section 2.2.4 as suggested in its section title in line 242.
COMPLITLY REVISED
- Does it make sense to remove the reference to Figure 10 in line 280?
CORRECTED
- Does it make sense to update “Fig. 1B” in line 298 to Figure 10B?
DONE
- Concerning the sentence starting in line 303, differences between direct and indirect ELISA are commonly noticed as requiring a secondary antibody to amply the signal or not, instead based on the molecule to be detected. See, for example, https://www.bio-rad-antibodies.com/elisa-types-direct-indirect-sandwich-competition-elisa-formats.html. Kindly amend this sentence to avoid any confusion.
REVISED
- I am not sure what the bold font in Section 3.1 refers to
CORRECTED.
- Concerning the sentence in line 312 and further to the point (13) as made above, I am not sure multi-valency will help to amplify the signal. I understand multi-valency means one binder (i.e., the multivalent phage) binds to multiple copies of the target (e.g., PSA), and further, the signal-generating moiety should be with the binder. Accordingly, multiple copies of the target binding to one binder will activate only one signal-generation moiety. Will it actually considered the opposite of signal amplification? Or, do we intend to use the phage as the capturing moiety and use another labeled antibody to generate the signal? If this is the case, will a nanoparticle carrying or a plate coated with the same peptide as the phage work similarly? Also, if this is the case, is it the reason why this method can distinguish free PSA since PSA expressed on cells after binding to a phage most likely hindered from binding to another antibody? In other words, I would really appreciate some summary on why, after the binder screening, the phage is still critical to have as a part of the detection method. Along this line, some examples of phage detection methods approved by a regulatory authority, such as FDA, would be really helpful, even if they may not relate to PC.
THE ESSENSE OF LANDSCAPE PHAGE WAS DESCRIBED IN MORE DETAIL
(19) Further to the sentence in line 315, I would really appreciate some discussions on why and how phage can distinguish immobilized f-PSA from immobilized but not free PSA. Are the binding sites in the PSA different?
THIS IS A KEY QUESTION, AND I HOPE TO RESPOND TO IT AFTER FINISHING MOLECULAR MODELIN OF PHAGE-PSA COMPLEXES..
After reading the current manuscript, I feel it seems premature to claim the use of phage displays for early diagnosis of prostate cancer.
For example, after arriving at a peptide, an antibody, a Fab, or a scFv from a phage display screening, is it still advantageous to use them when they are displayed on phages for diagnosis?
If the multi-valence of a landscape phage is critical,
THE CONCLUSION WAS COMPLETLE REVISED
will coated gold nanoparticles serve as a similar (if not better) detection tool, such as the commercially available COVID test kits?
DISCUSSED IN THE REVISED CONCLUSION
For those electrochemical immunosensors, will they still require expensive equipment and centralized laboratories?
THESE SENSORS ARE CONSIDED AS MODEL PLATFORMS FOR TESTING OF THE NEW INTERFACES , BUT NOT AS DEVICES FOR CLINICL USE
In addition, it seems the current draft does not provide any information linking PSA, t-PSA, f-PSA, or any of their ratios to PC patients in their early stages.
THE DATA ARE PROVIDED (TABLE 1)
Further, I do not see any discussion on the specificity and selectivity data of the selected peptides using patient samples, thus, it may be a little early to claim any clinical indications.
THE TITLE IS CHANGED
THE DATA ARE PROVIDED (TABLE 1)
THE CONCLUSION HAS BEEN REVISED.

Round 2
Reviewer 2 Report
Comments and Suggestions for Authors
Many thanks for the author’s kind amendments and patience. Now, it is clear to me that phage display is reviewed from the perspectives of uses as the diagnostic tool itself as well as screening assays to discover more prostate cancer probes. Yet, I do have the following remaining concerns, and would really appreciate some further clarification. See more details below.
The sentence starting in line 92 is confusing. Would the author mind some elaboration on the interactions among the peptides expressed on a landscape page display? What exactly are they? How would they be similar to, for example, CDR1, CDR2, and CDR3 in one heavy chain of an antibody? My understanding is that increased valency will increase the overall affinity. Yet, I am not sure if this sentence hints at other mechanisms behind the high affinity or other advantages mentioned. Also, if it makes sense, it is suggested to remove the sentences starting in lines 106 and 108 (or expanding them a little).
Concerning the conclusion starting in line 233, it seems there is still no evidence provided to support f-PSA/t-PSA ratio instead of t-PSA works better in diagnoses.
Is it possible to remove Figure 2? If it provides/supports something additional other than this Fab does bind to PSA, would the author mind highlighting it?
It still seems to me that the sentence starting in line 71 will fit better in other places, such as right before Section 2.1.
What does the “bispecific” in line 122 refer to? Shall we amend it to bivalent?
Usually, a sandwich ELISA is considered different from a direct ELISA. Accordingly, the phase spanning lines 308-309, as well as the “direct ELISA” in line 312, is a little confusing. Also, some quick definition on “landscape phage-based signaling antibody” would be really appreciated.
In addition, some minor formality issues were noticed, such as “,,” in line 48; “..” in lines 77, 152, 187, 289, and 362; “.,” in line 366; a missing “.” in line 83; “For example.” in line 208; “increasedr” in line 297; “Sselected” in line 325; “, Specifically,” in line 346; “constructionof” in line 349; “canrecognize” in line 381; “the development the new approaches for development of” in lines 294-295; “for for” in line 395; “n this contest” in line 396; and “. in contrast” in line 400. It is also suggested to provide full names of acronyms upon their first appearance and to use the acronyms from there. See, for example, the introduction of ELISA in lines 60, 290 and 307, as well as the introduction of mAb in lines 73, 203, and 292.
Author Response
Comments and Suggestions for Authors
Dear Reviewer, thank you very much for your great assistance in improving the Review. My responses are presented below.
The sentence starting in line 92 is confusing. Would the author mind some elaboration on the interactions among the peptides expressed on a landscape page display?
What exactly are they? How would they be similar to, for example, CDR1, CDR2, and CDR3 in one heavy chain of an antibody? My understanding is that increased valency will increase the overall affinity. Yet, I am not sure if this sentence hints at other mechanisms behind the high affinity or other advantages mentioned.
YOU ARE RIGHT. I ADDED A CITATION TO THE PAPER [142], WHICH REVIEWES ADVANTAGES OF CONSTRAINED DISPLAYED PROTEINS, NOT ONLY ANTIBOIES SCAFOLDS.
Also, if it makes sense, it is suggested to remove the sentences starting in lines 106 and 108 (or expanding them a little).
REMOVED
Concerning the conclusion starting in line 233, it seems there is still no evidence provided to support f-PSA/t-PSA ratio instead of t-PSA works better in diagnoses.
I AGREE AND ADDED THE REFERENCES TO PAPERS DISCUSSING THE ISSUE [69,144,145].
Is it possible to remove Figure 2? If it provides/supports something additional other than this Fab does bind to PSA, would the author mind highlighting it?
AGREE. THE FIGURE RENOVED.
It still seems to me that the sentence starting in line 71 will fit better in other places, such as right before Section 2.1.
AGREE. RELOCATED.
What does the “bispecific” in line 122 refer to? Shall we amend it to bivalent?
CORRECTED TO biospecific
Usually, a sandwich ELISA is considered different from a direct ELISA. Accordingly, the phase spanning lines 308-309, as well as the “direct ELISA” in line 312, is a little confusing. Also, some quick definition on “landscape phage-based signaling antibody” would be really appreciated.
THE REFERENCE [143] WAS ADDED, WHICH EXPLAIN THE PRINCIPLES AND TECHNIQUE OF SANDWICH PHAGE ELISA,
In addition, some minor formality issues were noticed, such as “,,” in line 48; “..” in lines 77, 152, 187, 289, and 362; “.,” in line 366; a missing “.” in line 83; “For example.” in line 208; “increasedr” in line 297; “Sselected” in line 325; “, Specifically,” in line 346; “constructionof” in line 349; “canrecognize” in line 381; “the development the new approaches for development of” in lines 294-295; “for for” in line 395; “n this contest” in line 396; and “. in contrast” in line 400. It is also suggested to provide full names of acronyms upon their first appearance and to use the acronyms from there. See, for example, the introduction of ELISA in lines 60, 290 and 307, as well as the introduction of mAb in lines 73, 203, and 292.
CORRECTED
